# Phytochemistry and Pharmacological Activity of Plants of Genus *Curculigo*: An Updated Review Since 2013

**DOI:** 10.3390/molecules26113396

**Published:** 2021-06-03

**Authors:** Ying Wang, Junlong Li, Ning Li

**Affiliations:** Inflammation and Immune Mediated Diseases Laboratory of Anhui Province, School of Pharmacy, Anhui Medical University, Hefei 230032, China; m18326039565@163.com (Y.W.); lijunlong3317@gmail.com (J.L.)

**Keywords:** genus *Curculigo*, phytochemistry, pharmacological activities, herbal medicine

## Abstract

The genus *Curculigo*, as a folk herbal medicine, has been used for many years in China, treating impotence, limb limpness, and arthritis of the lumbar and knee joints. The last systematic review of the genus *Curculigo* was written in 2013, scientifically categorizing the phytochemistry and biological activities. Hitherto, the original compounds and their pharmacological activities were presented as the development of this genus, but there is not an updated review. To conclude the progression of the genus *Curculigo*, we collected the new literature published from 2013 to 2021 in PubMed, Web of Science, Google Scholar databases, and the Chinese National Knowledge Infrastructure. The novel chlorophenolic glucosides, curculigine, phenolic glycosides, orcinosides and polysaccharides were isolated from *Curculigo*. The new analyzing methods were established to control the quality of *Curculigo* as a herbal medicine. In addition, the pharmacological effects of *Curculigo* focused on anti-diabetes, antibacterial, anti-inflammatory, osteoporosis, antioxidation, etc. The antitumor and neuroprotective activities were newly explored in recent years. The application of herbal medicine was gradually developed in scientific methods. The medicinal value of the genus *Curculigo* needs to further investigate its pharmacological mechanisms. This new review offers more insights into the exploitation of the pharmacological value of the genus *Curculigo*.

## 1. Introduction

Hypoxidaceae has about nine genera and 200 species, mainly distributed in the southern hemisphere of the Old World and North America. It is a small family of herbaceous perennial monocotyledons. These include leaves, roots or basal, usually with conspicuous parallel veins or folded fan-shaped veins; peanuts on flower stems which have petals in multiples of three, actinomorphic; fruit in capsules or pseudo-berries; plants with tubers or bulbs. *Curculigo* Gaertner is a small genus in the family of Hypoxidaceae. It includes 17 species and four varieties, and seven of these species (two endemic) are in China [1,2]. *Curculigo* species are perennial herbs, often with tuberous rhizomes [3]. Some species of genus *Curculigo* are recorded in traditional or fork herbalism worldwide for their medicinal properties. *Curculigo orchioides* is native to India and is found everywhere, from sea level to 2300 m above sea level, especially in rocky areas [4]. In China, *Curculigo orchioides* has a long history of medicinal use, and its rhizome was considered as a tonic medicine to maintain health energy and nourish the liver and kidney since the Tang Dynasty. *C. orchioides* was often applied for the treatment of impotence, limb limpness, arthritis of the lumbar and knee joints, and watery diarrhea [5]. In the Ayurvedic Medical System, *C. orchioides* was applied to treat jaundice, asthma, and urinary and skin diseases [6]. *Curculigo capitulata* was used as an ethnomedicine to treat the disorders of chronic bronchitis, nephritis and reproductive system inflammation in Dai medicine, Yunnan China [7]. *Curculigo breviscapa* is used to treat edema in the Guangxi province of China [8]. *Curculigo pilosa* was the first described aboriginal African species in the *Curculigo* genus, and it is commonly used as a purgative as well as for the management and treatment of hernias, infertility, diabetes, genital infections and sexually transmitted infections in the Yoruba tribe of South-West Nigeria [9]. *Curculigo recurvata* was used to treat snake bites and arthropod stings in the Congo [10].

The plants of the genus *Curculigo* contain diverse secondary metabolites with a variety of bio-activities. Up to now, 10 species including *C. orchioides*, *C. capitulata*, *C. sinensis*, *C. crassifolia*, *C. breviscapa*, *C. gracilis*, *C. recurvata*, *C. glabrescens*, *C. pilosa*, and *C. latifolia* have been chemically and pharmacologically investigated. Previous studies have revealed that phenols and phenolic glucoside, norlignans, and terpenoids were the main consituents in the *Curculigo* species [11]. These compounds and extracts of *Curculigo* plants exhibited various bio-activities including adaptive [12], immunomodulation [13], radical scavenging and anti-oxidation [14,15], sweet-tasting and taste-modifying, anti-inflammation [16], aphrodisiac, estrogenic and sexual behavior-modifying [17], neuroprotection [18], antidepression [19,20,21], nephroprotection [22], anti-osteoporotic [23,24,25], antiarthritic [26], anti-tumor [27], mast cell stabilization and antihistaminic [28], and antidiabetic activity [29].

The isolation, structures and bio-activities of the compounds from species belonging to genus *Curculigo* have been previously reviewed. These monographs [11,30,31] presented the traditional use, phytochemistry, pharmacology and toxicology of plants in the genus *Curculigo*. However, the phytochemicals of the medicinal plants of genus *Curculigo* are inadequately reviewed and the related pharmacological advances are not updated. It was only in 2020 that Pradeep Goyal et al. reviewed the morphology, pharmacology and chemical properties of *Curculigo orchioides* [32]. This current review comprehensively summarizes the latest literature dealing with the isolation, chemical structural elucidation, synthesis and pharmacological activities of the medicinal plants of the genus *Curculigo*, concentrating on the work that has appeared in the literature from 2013 up to March 2021.

## 2. Phytochemistry

The plants of the genus *Curculigo* contain diverse secondary metabolites with various biological activities. From 2013 to now, a number of chemical constituents have been isolated from the rhizomes or fruits of the plants of the genus *Curculigo*. The structure of the monomer compounds was identified by extensive spectroscopic analysis (UV, IR, HRESIMS, 1D and 2DNMR) and single-crystal X-ray diffraction analysis. The constituents of polysaccharides were determined by an ultraviolet-visible spectrophotometer (UV-2550, Shimadzu, Kyoto, Japan). As literature reported, polysaccharides, norlignans, phenolics, and terpenoids are the main bio-active compounds recently found in this genus. The isolation, characterization, chemistry, as well as related bioactivities of these compounds were highlighted in their chemical categorization.

### 2.1. Polysaccharides and Monosaccharide

From the rhizomes of *C. orchioides*, Wang et al. (2017) obtained a water soluble *O*-acetyl-glucomannan COP90-1 (**1**) with molecular weight 4609 Da, which was composed of (1→4)-linked β-d-Manp, (1→4)-linked 3-*O*-acetyl-β-d-Manp, and (1→4)-linked β-d-Glcp, (1,3→6)-linked β-d-Manp, and terminated with β-d-Manp and α-d-Glcp. SEM and Congo red analysis showed that COP90-1 was homogeneous with an inerratic ellipsoid shape instead of triple-helix structure. In vitro, COP90-1 effectively promoted the proliferation and differentiation of primary osteoblasts [33]. Two years later, Wang et al. (2019) isolated and purified a novel homogeneous heteropolysaccharide COP70-3 (**2**) with a main backbone chain of (1→5)-linked α-l-Araf, (1,3→5)-linked α-l-Araf, (1→6)-linked β-d-Galp, (1→4)-linked β-d-Manp, (1,2→5)-linked α-l-Araf, (1→3)-linked β-l-Rhap, (1,3→6)-linked β-d-Manp, (1→3)-linked α-d-GalpA, (1,3→6)-linked β-d-Galp and (1→6)-linked α-d-Glcp residues. Furthermore, in vitro COP70-3 displayed anti-osteoporosis activity indicated by obvious promotion of the differentiation of MC3T3-E1 cells at 1.87 nM, and significantly improved the osteogenic mineralization rate at 0.94 nM and 1.87 nM, respectively [34]. Chao Niu et al. used preparative reversed-phase high performance liquid chromatography (MeOH-H_2_O, 10:1~1:1, flow rate 2.5 mL beat min, wavelength 203 min) to obtain levoglucosan (**3**) [35] (Figure 1). Levoglucosan were obtained from the genus *Curculigo* for the first time.

### 2.2. Norlignans

Norlignans are typical compounds in the plants of section Molineria in the genus *Curculigo* [15,36,37,38]. From the roots of *Curculigo capitulate*, Li et al. (2019) identified three unprecedented 9-norlignans (capitulactones A-C, **4**–**6**) featuring a unique 3,5-dihydrofuro[2,3-d]oxepin-7(2H)-one scaffold. Their structures with the absolute configurations were unambiguously established by a combination of spectroscopic data, ECD analysis, and total synthesis. The unique scaffold of the common western hemisphere of the molecules was constructed by using the oxidation-reduction strategy from benzodihydrofuran (Figure 2) [39]. In addition, hypothetical biosynthetic pathways for capitulactones A–C (**4**–**6**) were also proposed (Scheme 1). Sinenside A (**7**) was a norlignan isolated from *Curculigo sinensis* characterized by a unique cyclic disaccharide (or saccharide dianhydride) in which two pyranose residues are fused with a challenging 1, 2-trans-configuration [38]. Paresh and Chepuri (2015) accomplished the total synthesis of sinenside A (**7**) in nine steps by using an intramolecular acetalization as the key step directly affords the target parent tricyclic ring system (Scheme 2) [40].

### 2.3. Chlorophenolic Glucosides

Curculigines are rare natural chlorine-containing compounds occurring in *C. orchioides*. Xu and his co-workers reported three chlorophenolic glucosides, B and C from the rhizome of *C. orchioides* in 1987 and 1992 [41,42]; Cao et al. (2009) identified curculigine D [43]. From rhizomes of *C. orchioides*, Wang et al. (2013; 2014; 2018) obtained eleven chlorophenolic glucosides as curculigine E–O (**8**–**18**). Among them, compounds **8**–**10**, **12**, **14**–**18** showed moderate effects on osteoblast proliferation with a proliferation rate of 10.1%–15.9% in MC3T3-E1 cells. Further structure–activity relationship analysis revealed that anti-osteoporosis activity of chlorophenol glycosides is higher than that of phenolic glycosides. The C-6-linked chlorine in chlorophenol glycosides may reduce their anti-osteoporosis activity and the introduction of a C-5 hydroxyl group into the aglycone of chlorophenol glycosides increases their anti-osteoporosis activity [44,45,46]. Deng et al. (2020) isolated two new chlorophenolic glucosides named curculigine P and Q (**19**, **20**) from the rhizomes of *C. orchioides*, and compound **19** displayed high inhibitory potential against 5-reductase in a HaCaT-based bioassay [47]. Chen et al. (2017) purified five chlorophenolic glucosides (**21**–**25**) from *C. orchioides*, and all the compounds showed no xanthine oxidase inhibitory activities [48]. In addition, curculigine A (**26**) was isolated from the same plant and showed no toxicity to human HL-7702 hepatocytes [49]. Chao Niu et al. isolated Orcinol-1-*O*-β-d-apiofuranosyl-(1→6)-β-d-glucopyranoside (**27**) and Anacardoside (**28**) by CC (ODS, MeOH-H_2_O, 1:5, MeOH-H_2_O, 1:1) [35] (Figure 3).

### 2.4. Phenolic Compounds

Phenolic compounds are the main metabolites occurring in the plants of the genus *Curculigo*. From the rhizomes of *C. orchioides*, Wang et al. (2013; 2014) obtained two new phenolic glucosides, orcinoside H (**29**) and curculigoside I (**42**), together with ten known ones as orcinol glucoside (**30**), orcinol glucoside B (**31**), curculigoside A (**32**), curculigoside B (**33**), curculigoside C (**34**), curculigoside G (**35**), 3-hydroxy-5-methylphenol-1-*O*-[β-glucopyranosyl-(1-6)-β-d-glucopyranoside] (**36**), 3-hydrox-5-methyl-phenol-1-*O*-[β-apiosyl-(1-6)-β-glucopyranoside] (**37**), glucosyringic acid (**39**), and benzyl-*O*-β-d-glucopyranoside (**40**). Among them, compounds **30**, **31**, **34**, **39**, and **42** showed weak anti-osteoporotic activity [44,45]. Jiao et al. (2013) isolated orcinol glucoside (**30**), curculigoside B (**33**) and glucosyringic acid (**39**) from rhizomes of *C. orchioides*, and all of these three have no hepatotoxicity against the human hepatic cell line HL-7702 [49]. Chen et al. (2017) reported two new heterocyclic phenolic derivatives, orcinosides I and J (**44** and **45**), together with four known phenolic compounds as 3-(4-hydroxy-3-methoxy-phenyl) propane-1,2-diol (**46**), 3-(4-Hydroxy-3,5-dimeth- oxyphenyl) propane-1,2-diol (**47**), piperoside (**48**), 4-ally-2, and 6-dimeoxy phenol glucoside (**49**). Among them, orcinosides I (**44**) and J (**45**) exhibited xanthine oxidase inhibitory activities with the inhibitor rates of 72.40 ± 4.47% and 36.71 ± 3.67% at the concentrations of 0.55 mM. The IC_50_ of **44** and **45** were 0.25 and 0.62 mM, respectively (Figure 4) [48]. The other two phenolic compounds, named *p*-Hydroxycinnamic acid (**51**) and 3,5-Dihydroxy-4-methoxybenzoic acid (**50**) were also isolated from dried and powdered rhizomes of curculigo by Chao Niu et al. In addition, the phenolic compounds were also isolated from dried and powdered curcuma rhizomes by Chao Niu et al. [35]. Compounds **50** and **51** were obtained from the genus *Curculigo* for the first time.

### 2.5. Terpenoids

Terpenoids are a small class of second metabolites found in *Curculigo* species. From the rhizomes of *C. orchioides*, a new cycloartane-type triterpenoid ketone (**52**) was isolated [49], and **52** showed hepatotoxicity against the human hepatic cell line HL-7702 in vitro. Zhang et al. (2019) obtained six terpenoids from *C. orchioides* as (3S,5R,6S,7E,9R)-megatigma-7-ene-3,5,6,9-tetrol (**53**), actinidioionoside (**54**), (6S,9R)-roseoside (**55**), (−)-angelicoidenol-2-*O*-β-d-glucopyranoside (**56**), (−)-angelicoidenol-2-*O*-β-apiofuranosyl-(1→6)-β-d-gluco-pyranoside (**57**), tetillapyrone [(7R,9S,10R)-3-methyl-5-(4-hydroxyl-5- hydroxylmethyl-tetrahydrofuryl)]-6-hydroxypyran-2-one (**58**) (Figure 5) [25].

### 2.6. Cyclic Peptides

Besides the above compounds, Chen et al. (2017) yielded eight known cyclodipeotides as cyclo-(l-Ala-l-Tyr) (**59**), cyclo-(l-Ser-l-Phe) (**60**), cyclo-(LeuAla) (**61**), cyclo-(Leu-Thr) (**62**), cyclo-(Leu-Ser) (**63**), cyclo-(S-Pro-R-Leu) (**64**), cyclo-(Val-Ala) (**65**) and cyclo-(Gly-d-Val) (**66**) were obtained from the rhizomes of *C. orchioides* (Figure 6) [48].

## 3. Pharmacological Properties

Since 2013, there have been a lot of studies and explorations on the pharmacological effects of the genus *Curculigo*, which have been classified into 12 parts: Anti-Diabetic Activity, Anti-Osteoporosis, Antioxidant, Neuroprotective Effect, Antitumor, Antibacteria, Anti-Inflammation and Anti-Arthritis, Anti-Diarrhea and Anti-Nociception, Effect on Perimenopausal Syndrome, Male Reproductive Improvement, Cardio-Protection, Other Activities. All of this was summarized in “Table 1”.

### 3.1. Anti-Diabetic Activity

Karigidi et al. (2020) reported that *Curculigo pilosa* supplemented (CPS) diet and significantly lowered the blood glucose in streptozotocin (STZ) induced diabetic rats. A CPS-diet significantly enhanced the activities of hepatic glycolytic and lowered the gluconeogenic enzymes in diabetic animals. A CPS-diet also restored the lipid profile, oxidative stress markers and serum markers of hepatic and renal damages in the diabetic rats [50]. In another experiment conducted by Karigidi and Olaiya (2020), oral administration of an extract of *C. pilosa* effectively mitigated the hyperglycemia mediated oxidative damage via improving the antioxidant system to inhibit the generation of lipid peroxide, hydrogen and nitric oxide [51]. Moreover, an extract of *Curculigo pilosa* also demonstrated a strong ability to inhibit biomarker enzymes [52]. The fruits and root aqueous extract of *Curculigo latifolia* exerted antidiabetic and hypolipidemic effects through altering regulation genes in glucose and lipid metabolisms in STZ-induced diabetic rats, indicated by an increase in body weight, high density lipoprotein (HDL), insulin, and adiponectin levels, and a decrease in glucose, total cholesterol (TC), triglycerides (TG), low density lipoprotein (LDL), urea, creatinine, ALT, and GGT levels [53]. Singla et al. (2020) found that hydroalcoholic and ethanolic extracts of *C. orchiodies* at doses of 150, 300 and 600 mg/kg significantly attenuated a hyperglycemia induced increase in lipid profile, oxidative stress and normalized the damaged renal functions (albumin, urea and creatinine) in STZ-nicotinamide induced diabetic nephropathy rats [22]. The study by Gulati et al. (2015) showed that ethanolic extract of *C. orchioides* exerted anti-diabetic activity and enhanced glucose uptake in murine 3T3-L1 adipocytes, and inhibited adipogenesis in a cell-based assay [54].

### 3.2. Anti-Osteoporosis

Osteoporosis is a common bone disease characterized by increasing osseous fragility and fracture due to the reduced boss mass and micro-structural degradation. The main strategies for osteoporosis are hormone replacement treatment (HRT) and alendronate therapies that may produce adverse side-effects [55]. The rhizome of *C. orchioides* and related traditional Chinese medicine formulas with strengthening of tendons and bone functions, such as “Er-Xian-Tang”, were widely used to prevent and treat osteoporosis [56]. Network pharmacology analysis of active components in *C. orchioides* targeting osteoporosis predicted that curculigoside (**32**) and other compounds had therapy potentials for osteoporosis [57,58,59]. Shen et al. (2013) reported that 100 μM curculigoside significantly enhanced the proliferation of bone marrow stromal cells (BMSCs), increased the osteogenic gene expression, and profoundly up-regulated the secretion of osteoprotegerin (OPG) [60]. In the study by Liu et al. (2014), curculigoside (**32**) from *C. orchioides* stimulated alkaline phosphatase (ALP) activity and calcium deposition of human amniotic fluid-derived stem cells (hAFSCs) during osteogenic differentiation in a dose-dependent manner (1–100 μg/mL), and increased the gene expression of osteogenic osteopontin (OPN) and collagen I, inhibiting the osteoclastogenesis through the Wnt/β-catenin signal transduction pathway [23]. Zhu et al. (2015) reported that curculigoside (**32**) was reported to exert anti-osteoporosis activity via regulated proliferation, differentiation and inhibiting of the levels of pro-inflammatory cytokines TNF-α, IL-1β, IL-6 and COX-2 in dexamethasone-induced rat calvarial osteoblasts [16]. In vivo, curculigoside can also promote the calcium deposition of osteoblasts under oxidative stress, increase the levels of ALP and Runx2, and reduce the production and deposition of Aβ-induced osteoclasts through antioxidation in APP/PS1 mutated mransgenic mice [61]. In the study conducted by Zhang et al. (2019), curculigoside (**32**) was reported to prevent excess-iron-induced bone loss in mice and osteoblastic MC3T3-E1 cells through anti-oxidation and inhibiting excess-iron- induced phosphorylation of the Akt-FoxO1 pathway that target the genes of MnSOD, Gadd45a, Bim, FasL, and Rab7 [24].

Beside the curculigoside (**32**), other compounds with anti-osteoporosis obtained from the plants of genus *Curculigo* activity were also documented. From the rhizomes of *C. orchioides,* Wang et al. (2017, 2019) isolated two polysaccharides as COP90-1 (**1**) and COP70-3 (**2**), and both **1** and **2** displayed anti-osteoporosis activities in vitro [33,34]. Wang et al. (2013; 2014; 2018) reported that nine chlorophenolic glucosides, curculigine E-G, I, and K–O (**8**–**10**, **12** and **14**–**18**), displayed anti-osteoporosis by increasing the proliferation rate by 10.1%–15.9% in MC3T3-E1 cells [44,45,46].

### 3.3. Antioxidant

Excessive oxidative stress may cause severe damage to proteins, lipids, DNA, and may even result in cell death and trigger many diseases, such as cancer, cardiovascular disease, diabetes mellitus, and the aging process, etc. [62]. The ethanol extract of *C. orchioides* rhizomes (1–25 μg/mL) showed the high antioxidant activity on superoxide anion radicals, hydroxyl radicals, hydrogen peroxide and DPPH free radicals, and lipid peroxidation in a concentration-dependent manner, and protected from cisplatin-induced auditory damage by inhibiting lipid peroxidation and scavenging activities against free radicals at a concentration range of 2.5–25 μg/mL [63]. Phytochemical profile analysis revealed that orcinol-β-d-glucoside (**30**) and curculigoside A (**32**) were the main antioxidant compounds in *C. orchioides* [64]. Murali and Kuttan (2015) reported that methanolic extract of *C. orchioides* enhanced the cancer chemotherapeutic efficacy of cyclophosphamide in mice, and had ameliorative effects on cyclophosphamide-induced oxidative stress by lowering the level of lipid peroxidation [65]. The extract of *C. orchioides* restored levels of anti-oxidant enzymes of SOD, CAT, and GST, and significantly decreased the lipid peroxidation in the rat liver induced by Chromium (VI) treatment [66]. In the experiments performed by Hejazi et al. (2018), the ethylacetate fraction (EA) of *C. orchioides* exhibited significant radical scavenging activity on a DPPH assay with the IC_50_ as 52.93 μg/mL; *C. orchioides* also refurbished the anti-oxidant enzyme defense of mammalian tissue during oxidative stress. EA and aqueous ethylacetate (AEA) fraction effectively inhibited the growth of tumor cell lines of HepG2, HeLa and MCF-7, down-regulated the levels of antiapoptotic Bcl-2 expression, and up-regulated the expression of apoptotic proteins caspase-3 and caspase-8 through an intrinsic ROS-mediated mitochondrial dysfunction pathway [67].

*Curculigo pilosa*, also called Africa crocus, is a folklore diet-medicine in Nigeria. The aqueous extract of *C. pilosa* seeds significantly inhibit the lipid peroxidation of rat penile homogenate induced by FeSO4 and SNP [68]. The extract of *C. pilosa* effectively mitigated the hyperglycemia mediated oxidative damage via improving the antioxidant system to inhibit the generation of lipid peroxide, hydrogen and nitric oxide in STZ-induced diabetic rats [51]. Ruzu herbal bitters (RHB), an anti-obesity medicinal concoction in Nigeria, mainly made from the root of *C. pilosa**,* significantly increased the activity of peroxidase, CAT and GSH in the liver and brain in high-fat diet induced diabetic rats [69]. For *C. latifolia*, the extract of the tuber displayed higher antioxidant activity than that of a leaf in a DPPH scavenging assay and a superoxide dismutase assay [70]. The subcritical water extraction of *C. latifolia* roots displayed significant antioxidant activity in DPPH and ABTS assays, and further LC-MS analysis revealed that phenolic compounds such as monobenzone, hydroquinone, phloridzin, pomiferin, mundulone, scandenin, and dimethyl caffeic acid were the main active compounds [71].

### 3.4. Neuroprotective Effect

The extract of *C. orchioides* rhizome exhibited significant inhibitory effects on acetylcholinesterase, which suggested the potential of *C. orchioides* for the treatment of Alzheimer’s disease [72] G.K. Pratap et al. used bio-autograph and spectrophotometry to determine the anti-acetylcholinesterase activity of methanol extracts of *C. orchioides* in vitro. The experiment showed that the ethanol extract had a good inhibitory effect on the AChE enzyme. This may help to explain the role of methanol extract from rhizome as an anti-acetylcholinesterase drug in the treatment of Alzheimer’s disease [73]. The curculigoside (**32**) in *C. orchioides* can effectively improve the behavior of Alzheimer’s rats, reduce the damage of hippocampal neurons, and inhibit the apoptosis of hippocampal neurons [74]. Dipica Ramchandani et al. induced neurotoxicity in mice with cyclophosphamide. According to the experimental results, it was inferred that the neuroprotective effect of *C. orchioides* may be due to flavonoids and polyphenols [75]. Recently, the antidepressant effects of curculigoside were reported, and related research revealed that curculigoside (**32**) improved depression-like behavior in mice, and significantly up-regulated the levels of BDNF protein, as well as the concentrations of DA, NE and 5-HT in the hippocampus of depressive animals [76,77]. In vitro, curculigoside A showed the potential for neurovascular repair therapy for stroke and brain injury by modulating the angiogenesis in cerebral endothelial cells via VCAM-1/Egr-3/CREB/VEGF signaling [78]. In vivo, curculigoside A alleviated the cerebral ischemia and reperfusion injury in the middle cerebral artery occluded (MCAO) model rats, induced the cell proliferation and angiogenesis through the Wnt5a/β-catenin and VEGF/CREB /Egr-3/VCAM-1 signaling axis, and promoted the maturation and stability of new blood vessels via increasing Ang1 and Tie-2 expression [79]. In addition, curculigoside exhibited evident neuro-protective effects against N-methyl-D-aspartate (NMDA) induced neuronal excitoxicity by prevented NMDA-induced neuronal cell loss, and reduced the number of apoptotic and necrotic cells [80].

Ge et al. (2014) described that orcinol glucoside (OG, **30**) improved the depressive behavior of CUMS rats by down-regulating the over-activity of HPA axis and increasing the expression of BDNF and phosphorylation of ERK1/2 in the hippocampus [19]. In another study performed by Wang et al. (2016), it was revealed that OG effectively alleviated anxiety-like behaviors in mice in three different tests, while it had no sedative effects on the animals [76].

Crassifoside H (CH) is a chlorine-containing norlignan obtained from *Curculigo glabrescens*. Zhang et al. (2017) reported that CH effectively improves the depression-like behavior of CUMS rats, improved HPA axis hyperactivity, reduced plasma CORT and CRH expression in hypothalamus, reversed CUMS-induced decrease of 5-HT1A receptor expression, and up-regulated BDNF and phosphorylated-ERK1/2 levels in the hippocampus [21]. Recent work by Li et al. (2020) revealed that CH not only alleviated the HPA axis dysfunction in CUMS-induced depressive mice, but also the suppressed expression of inflammatory cytokines of TNF-α and IL-1β, as well as the inhibition of NLRP3 inflammasome activation in the hippocampus [81].

### 3.5. Antitumor

Polysaccharides from *C. orchioides* displayed obvious anti-tumor effects on cervical cancer in mice, and significantly enhanced immune function and induction of apoptosis, indicated by an increase in the thymus and spleen index. Moreover, the polysaccharides significantly up-regulated the expression of caspase-3, caspase-9 and p53 protein in HeLa cells in vitro [27]. The silver nanoparticles (CoBAgNPs) prepared from *C. orchioides* rhizome extracts in vitro showed an inhibitory effect against human breast cancer cell lines (MDA-MB-231) and Vero cell lines with the IC_50_ values as 18.86 μg/mL and 42.43 μg/mL, respectively [82]. In another study conducted by Selvaraj and Agastian (2017), the ethyl acetate extract and silver nanoparticles (80 μg/mL) synthesized from *C. orchioides* exhibited maximum growth inhibitory effects against human breast cancer cells (MCF-7) with a rate of 66.12% and 71.28% [83].

Murali and Kuttan (2016) reported that the phenolic glucoside curculigoside (**32**) enhanced the natural killer (NK) cell activity, antibody-dependent cell-mediated cytotoxicity and complement-mediated cytotoxicity in B16F10 melanoma cancer-cell-bearing mice [84]. In addition, compound **32** also significantly increased the levels of TH1 cytokines IL-2 and IFN-γ, reduced the formation of pulmonary metastatic colonies and prolonged the life span of experimental animals. Orcinol glucoside (OG, **30**) is another main phenolic glucoside in the rhizomes of *C. orchioides*. Nahak et al. (2018) reported that OG loaded nanostructured lipid carriers (NLC) coated with PEG-25/55-SA exhibited enhanced anticancer activity in gastric, colorectal and hepatocellular carcinoma in vitro [85].

Xiaoyu Wang et al. introduced that Erxian decoction, a traditional Chinese medicine containing *C. orchioides*, can inhibit the metastasis and invasion of ovarian cancer in vivo and in vitro through EGFR, ErbB, MMP2, MMP7, MMP9 and VEGFR [86].

### 3.6. Antibacteria

Marasini et al. (2015) reported that the alcohol extract of *C. orchioides* had high antibacterial activity against methicillin-resistant *Pseudomonas aeruginosa,* with the MIC value as 49 μg/mL, whereas it had no effect against gram-negative bacteria [87]. Perumal et al. (2017) prepared silver nanoparticles (AgNPs) from *C. orchioides* leaf extracts, and the phytochemical loaded-silver nanoparticles of *C. orchioides* showed good inhibitory activities against *P. aeruginosa* and *S. aureus*, with lower activities towards *E. coli* and *K. pneumonia* [88]. D. C. Nwokonkwo analyzed the phytochemical composition and antibacterial activities of fresh rhizome powder of *Curculigo pilosa*. The results showed that the minimum inhibitory concentrations ((MIC)) of crude ethanol extract and neutral metabolites were 25 mg/mL, 50 mg/mL and 100 mg/mL, respectively [89]. Mohammad Shah Hafez Kabir et al. found that the maximum inhibitory range of methanol extract from *C. recurvata* to *Bacillus cereus* was 10.50 ± 0.5 mm [90]. Tianyan Yun et al. found newly isolated extracts from medicinal plants (*Curculigo Capulata*) showed high antifungal activity against Foc TR4. At the same time, the optimization method of fermentation broth was established, and the bacteriostatic activity was increased by 72.13% [91].

### 3.7. Anti-Inflammation and Anti-Arthritis

Curculigoside A (**32**), the main active compound in *C. orchioides*, significantly relieved the hind paw swelling and arthritis index, reduced the serum pro-inflammatory factor levels of IL-6 IL-1β, PGE2 and TNF-α, decreased MDA and increased SOD activity in serum, and effectively down-regulated the expression of the NF-κB/NLRP3 pathway in Freund’s complete adjuvant (FCA) induced adjuvant arthritis rats [92]. In another arthritis rat model induced by type II collagen, compound **32** inhibited paw swelling and arthritis scores, decreased serum pro-inflammatory factor levels of TNF-α, IL-1β, IL-6, IL-10, IL-12 and IL-17A, down-regulated the expression of JAK1, JAK3 and STAT3, and up-regulated NF-κB p65 and IκB [26]. Moreover, the anti-arthritis activity of curculigoside A (**32**) was further confirmed by network pharmacological analysis [57].

### 3.8. Anti-Diarrhea and Anti-Nociception

Ahmad et al. (2020) reported that methanol extracts of *C. recurvate* had significant peripheral anti-nociception effects (400 mg/kg·bw) by inhibiting both early and late phases of nociception in the formalin-induced writhing test. In the castor oil-induced diarrhea model, *C. recurvate* extract significantly prolonged the onset time of diarrhea, reduced the volume and weight of intestinal contents in mice, and significantly decreased gastrointestinal motility. Further in silico molecular docking analysis showed that curculigine and isocurculigine possessed the highest affinity for COX-1 and COX-2; isocurculigine was identified as the most effective anti-diarrheal compound [93].

### 3.9. Effect on Perimenopausal Syndrome

The total glucosides of *C. orchioides* (TGC) showed a therapeutic effect on perimenopause model mice by increasing the index of thymus, uterus and spleen (TI, UI, SI), testosterone (T) and estradiol (E_2_), and reducing the level of luteinizing hormone (LH) [94,95]. Curculigoside improved the depression-like behaviors of the perimenopausal depression modeling mice in TST and FST experiments, increased the memory of mice, and reduced the number of times of electric shock and immobility. Curculigoside also significantly increased the organ index of the thymus, spleen and uterus, increased the levels of E2 and T in serum, increased the concentration of 5-HT and DA in brain tissue, decreased the levels of FSH and LH in serum, and improved the histopathological damage of the uterus, hypothalamus, thymus and spleen [95].

### 3.10. Male Reproductive Improvement

Blamus^TM^, a standardized extract produced from *C. orchioides*, significantly enhanced the free testosterone in serum in male rats at a dose of 50 mg/kg·bw. The structural integrities of seminiferous tubules, spermatogenesis, sperm morphology, interstitial cells and Sertoli cells in male rats were evidently improved after oral administration of Blamus^TM^ at the doses of 10, 25 or 50 mg/kg·bw [96,97].

### 3.11. Cardio-Protection

Zhao et al. (2020) reported that curculigoside (**32**) had a protective effect against myocardial ischemia-reperfusion injury (MIRI). In this study, curculigoside (**32**) significantly increased cell survival rate, and reduced mitochondrial-mediated cell apoptosis by inhibiting the opening of mitochondrial permeability transition pores (MPTP) [18].

### 3.12. Other Activities

Curculigoside A from the rhizome of *C. orchioides* improved the survival rate of random skin flaps in rats by inducing angiogenesis and alleviating ischemia-reperfusion (I/R) injury, increasing the VEGF and SOD expression and microvessel development and reducing the MDA level [98].

All of the pharmacological effects we mentioned in Part 3 were summarized in Table 1.

## 4. Conclusions

Most *Curculigo* species are used as folk or ethno-medicines in the regions of Asia and Africa. Among them, *C. orchioides* is a traditional medicinal herb with a long use history in Asian countries such as China, India, Nepal and Malaysia, etc. *Curculigo* plants have been reported to have diverse bioactive constituents including polysaccharides, norlignans, phenolics, and terpenoids. In this work, we provided an updated review on the phytochemistry and pharmacological activities of the medicinal plants of the genus *Curculigo,* from 2013 up to now. A total of 61 compounds have been isolated from the medicinal herbs of *Curculigo*, and it was revealed that phenolics were the main bioactive constituents in the plants of *Curculigo*. Recent modern pharmacological studies have demonstrated a variety of activities of the plants of *Curculigo* in vivo and in vitro, including anti-diabetes, anti-osteoporosis, anti-oxidation and lipid peroxidation inhibition, anti-depression, anti-arthritis, anti-nociception, anti-tumor, anti-bacteria, inhibition of ischemia-reperfusion injury, alleviation of perimenopausal syndrome and enhancement of male reproduction. It should be pointed out that *C. orchioides* is the species of most concerned in the genus *Curculigo*, and curculigoside and orcinol glucoside were the two main compounds in it.

Overall, herein, the updated advance on the medicinal plants of genus *Curculigo*, including the isolation, structural characterization, pharmacological activities, and relevant molecular mechanisms of the active compounds or extracts from medicinal plants of genus *Curculigo,* were summarized from 2013 up to now. The aim of this present review is to provide valuable references for researchers, and to promote the reasonable and scientific development and utilization of the medicinal resource of the genus *Curculigo*.

## Data Availability

Data sharing not applicable to this article as no datasets were generated or analysed during the current study. Our manuscript does not produce new data; all available data are contained in the non published material.

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
