# Peer review of "Phytochemistry and Pharmacological Activity of Plants of Genus Curculigo: An Updated Review Since 2013"

_molecules, 2021, doi:10.3390/molecules26113396_

Round 1
Reviewer 1 Report
The authors present an updating of literature relevant to the phytochemistry and pharmacological activity of plants from the genus Curguligo since 2013, when a very detailed publication (cited as ref 8) Medicinal plantsofgenus Curculigo: Traditional uses and a phytochemical and ethnopharmacological review by YanNie a,b,1, XinDong a,1, YongjingHe a, TingtingYuan a,b, TingHan a, KhalidRahman c,Luping Qin a,nn, QiaoyanZhang a appeared in the J of Ethnopharmacology. Nevertheless, their approach regarding pharmacology is rather superficial. They should better include a table including: Tested substances, Species, in vivo/in vitro, Model Administration (in vivo), Dose range, Active concentration, References and then devote a discussion to present pros and cons of their review. Toxicological implications should be also reported. The phytochemistry section is better presented but as I indicate in the attached file, if the figures have not been designed from scratch they have to ask for permission from published work. This is a crucial point for accepting the manuscript. Abstract must be written again, In the Introduction the starting date of the review (2013) should be better explained for the reader)

Author Response
- “Nevertheless, their approach regarding pharmacology is rather superficial. They should better include a table including: Tested substances, Species, in vivo/in vitro, Model Administration (in vivo), Dose range, Active concentration, References and then devote a discussion to present pros and cons of their review.Toxicological implications should be also reported.”
Response: Thank you for your professional opinion. Following the suggestion mentioned above, we have made a new table “Table 1” to systematically sum up the types of information about genus Curguligo you suggested. You can find it out in the new manuscript.
- “The phytochemistry section is better presented but as I indicate in the attached file, if the figures have not been designed from scratch they have to ask for permission from published work. This is a crucial point for accepting the manuscript.”
Response:Thank you for your worrying. All figures were originally designed by us, and the original ChemDraw files have been uploaded as the supplemental materials. We claimed that we have use rights of these figures.
- “Abstract must be written again.”
Response: The abstract have been written again, updating the new content that we revised in the new manuscript. Also, we reassured the grammar accuracy of the abstract. You can check it out in the new manuscript.
- “In the Introduction the starting date of the review (2013) should be better explained for the reader.”
Response: The starting date of our review were stated in line 72-73, the “Introduction” part.
Reviewer 2 Report
7-5-2021
I found the manuscript titled “phytochemistry and Pharmacological Activity of Plants of Genus Curculigo: An Updated Review from 2013” clear and well-structured but there are two important drawback point such as: originality and lack of information.
In order to improve the proposed work, a small list of papers not included in this review is provided below:
Mehta, J., & Nama, K. S. (2014). A review on ethanomedicines of Curculigo orchioides Gaertn (Kali Musli): Black Gold. Int J Phar & Biomedi Res, 1(1), 12-6.
GOYAL, P., & KABRA, A. (2020). A REVIEW ON PHYTOCHEMICAL AND PHARMACOLOGICAL PROFILE ON Curculigo orchioides. PLANT CELL BIOTECHNOLOGY AND MOLECULAR BIOLOGY, 243-252.
Ramchandani, D., Ganeshpurkar, A., Bansal, D., Karchuli, M. S., & Dubey, N. (2014). Protective effect of curculigo orchioides extract on cyclophosphamide-induced neurotoxicity in murine model. Toxicology international, 21(3), 232.
Yun, T., Zhang, M., Zhou, D., Jing, T., Zang, X., Qi, D., Xie, J. (2021). Anti-Foc RT4 Activity of a Newly Isolated Streptomyces sp. 5–10 From a Medicinal Plant (Curculigo capitulata). Frontiers in microbiology, 11, 3544.
VELMANI, S., MARUTHUPANDIAN, A., PERUMAL, B., & VIJI, M. (2019). MULTIPOTENTIAL MEDICINAL VALUE OF Curculigo orchioides Gaertn. Ethnomedicinal Plants with Therapeutic Properties, 21.
Pratap, G. K., Rather, S. A., & Shantaram, M. (2020). In Vitro Anti-cholinesterase Activity and Mass spectrometric Analysis of Curculigo orchioides Gaertn., Rhizome Extract. Analytical Chemistry Letters, 10(4), 442-458.
Kabir, M. S. H., Ahmad, S., Mahamoud, M. S., Chakrabarty, N., Hoque, M. A., Hossain, M. M., Shoibe, M. (2016). Comparative study of hypoglycemic and antibacterial activity of organic extracts of four Bangladeshi plants. Journal of Coastal Life Medicine, 4(3), 231-235.
Wang, X., Xu, L., Lao, Y., Zhang, H., & Xu, H. (2018). Natural products targeting EGFR signaling pathways as potential anticancer drugs. Current Protein and Peptide Science, 19(4), 380-388.
Zulfiqar, F., Khan, S. I., Ali, Z., Wang, Y. H., Ross, S. A., Viljoen, A. M., & Khan, I. A. (2020). Norlignan glucosides from Hypoxis hemerocallidea and their potential in vitro anti-inflammatory activity via inhibition of iNOS and NF-κB. Phytochemistry, 172, 112273.
Nwokonkwo, D. C. (2014). Antibacterial Susceptibility of the Constituents of Ethanol Crude Extract and the Neutral Metabolite of the Root of Curculigo pilosa Hypoxidaceae. International Journal of Chemistry, 6(4), 19.
Venkatachalam, P., Kayalvizhi, T., Udayabanu, J., Benelli, G., & Geetha, N. (2017). Enhanced antibacterial and cytotoxic activity of phytochemical loaded-silver nanoparticles using Curculigo orchioides leaf extracts with different extraction techniques. Journal of Cluster Science, 28(1), 607-619.
In addition, these statements need further explanation:
Abstact: How data were collected until 2020 or 2021?
Introduction: Given the partial diffusion of this plants genus, I believe that it would be appropriate to deepen the botanical aspect of Curculigo genus.
Introduction: please insert most current reference (i.e. 5, 7,)
Line 31: Following what reported by Govaerts, (2016) “The genus Curculigo includes 17 species and 4 varieties” and not around 20 as the authors report. Please revised your sentence or include a reference. Govaerts, (2016) World checklist of Hypoxidaceae. Facilitated by the Royal Botanic Gardens, Kew. Published on the internet: http:// apps.kew.org/wcsp/
Section 2: In my opinion, this section called “phytochemistry” is too difficult to read and it seems too technical to be a review.
- For example see the line 88-92.
- For example inside section 2 the authors deal some topics (i.e Polysaccharides, ….,Terpenoids) without explaining the role of newly identified compounds.
Author Response
- “The lack of information”
Response: We are very appreciated that you searched out these literature that we missed. In this revision, the content from 7 of 10 papers has been added into the new manuscript as the highlights in References. In these literature you listed, a study named “Enhanced antibacterial and cytotoxic activity of phytochemical loaded-silver nanoparticles using Curculigo orchioides leaf extracts with different extraction techniques.” has been written into our review as Reference No. 87. A study named “Norlignan glucosides from Hypoxis hemerocallidea and their potential in vitro anti-inflammatory activity via inhibition of iNOS and NF-κB” discussed the genus Hypoxis, which is belong to the family Hypoxidaceae, but it did not relate to genus Curculigo. In a review named “A REVIEW ON PHYTOCHEMICAL ANDPHARMACOLOGICAL PROFILE ON Curculigo orchioides” , the authors summarized the studies of species Curculigo orchioides. However, our review has fully covered the content of this review, so we did not add into our manuscript.
- “Abstact: How data were collected until 2020 or 2021?”
Response: The literature collection were ranged from 2013 to 2021. The collection date and methods have been explained in the abstract of the new manuscript.
- “Introduction: Given the partial diffusion of this plants genus, I believe that it would be appropriate to deepen the botanical aspect of Curculigo genus.”
Response: Thank you for your suggestion. We agreed that the botanical aspect is a important part of genus Curculigo. Also, we briefly described the botanical information in line 29-33, the “Introduction” part, and the aim of this review is to focus on the pharmacological value and phytochemistry of genus Curculigo.
- “Introduction: please insert most current reference (i.e. 5, 7,)”
Response: Reference No. 1, No.2 and No 4 are newly added into the “Introduction” part asline 28-35, 37-39.
- “Line 31: Following what reported by Govaerts, (2016) “The genus Curculigo includes 17 species and 4 varieties” and not around 20 as the authors report. Please revised your sentence or include a reference. Govaerts, (2016) World checklist of Hypoxidaceae. Facilitated by the Royal Botanic Gardens, Kew. Published on the internet: http:// apps.kew.org/wcsp/”
Response: Following your opinion, we has revised the inaccurate expression of this part, and we rewrite the sentence as line 28-35.
- “Section 2: In my opinion, this section called “phytochemistry” is too difficult to read and it seems too technical to be a review”
Response: It can not be denied that this part is not easy to read for non-professionals, but these expression would be more accurate to present the content from these literature.
Reviewer 3 Report
The manuscript describes the chemical components in Curculigo species and their pharmacological activities. However, the authors should cite the papers before 2013, because the genus is comparative small genus. In the manuscript, although the flavonoids which are very common compounds in plants are not reported, they may be known in the genus before 2013. So all compounds including the reports before 2013 should be described in the review. So I re-check the manuscript after revision.
Author Response
The manuscript describes the chemical components in Curculigo species and their pharmacological activities. However, the authors should cite the papers before 2013, because the genus is comparative small genus. In the manuscript, although the flavonoids which are very common compounds in plants are not reported, they may be known in the genus before 2013. So all compounds including the reports before 2013 should be described in the review. So I re-check the manuscript after revision.
Response: Thanks to your professional comment. The reason why we did not adopt the studies before 2013 is that there is a detailed review of genus Curculigoto summarize these content (Nie Y, Dong X, He Y, et al. Medicinal plants of genus Curculigo: traditional uses and a phytochemical and ethnopharmacological review. J Ethnopharmacol. 2013;147(3):547-563). And our manuscript is the update review to summarize the progressing work of genus Curculigo. Moreover, the former review emphatically described the phytochemistry work before 2013. In recent years, there were many studies to explore the pharmacological value of genus Curculigo which are main content in our review.
Reviewer 4 Report
The paper is focused on plants from the genus Curculigo showing the last information about phytochemistry and pharmacological activity of those plants. The review is well organized and relatively updated. There are few suggestions:
In the introduction, I would add a sentence about the family Hypoxidaceae.
As well as why this genus is important in the whole family?
The literature should be updated. There are many papers regarding pharmacological activity published in 2020 and 2021 which are not reviewed (Ex. Chemical Constituents of Curculigo orchioides By: Niu, Chao; Zhang, Zhen-Zhen; Yang, Lan-Ping; et al. CHEMISTRY OF NATURAL COMPOUNDS Volume: 56 Issue: 5 Pages: 957-959 Published: SEP 2020)
L 14 Which methods have been used to identify the compounds?
Author Response
- “In the introduction, I would add a sentence about the family Hypoxidaceae.As well as why this genus is important in the whole family?”
Response: Thank you for your opinion. We have added the related content into our manuscript, as line 28-35, 37-39, the “Introduction” part.
- “The literature should be updated. There are many papers regarding pharmacological activity published in 2020 and 2021 which are not reviewed (Ex. Chemical Constituents of Curculigo orchioides By:Niu, Chao; Zhang, Zhen-Zhen; Yang, Lan-Ping; et al. CHEMISTRY OF NATURAL COMPOUNDS Volume: 56 Issue: 5 Pages: 957-959 Published: SEP 2020)”
Response: We have updated again the literature we collected. And we claimed that data collection is ranged 2013 to 2021, as line 71-74, the “Introduction” part.
- “Which methods have been used to identify the compounds?”
Response: The methods has been added into line 78-82, the “Phytochemistry” part.
Round 2
Reviewer 2 Report
Authors modified the manuscript following the reviewer suggestions.
Author Response
Thank you for your positive report.
Reviewer 3 Report
I considered tha manuscript as accept.
Author Response
Thank you for your positive report.